# Improved Anomaly Detection by Using the Attention-Based Isolation Forest

Lev Utkin †, Andrey Ageev †, Andrei Konstantinov † and Vladimir Muliukha *,†

Higher School of Artificial Intelligence, Peter the Great St. Petersburg Polytechnic University, Polytechnicheskaya, 29, 195251 St. Petersburg, Russia
* Correspondence: vladimir.muliukha@spbstu.ru
† These authors contributed equally to this work.

**Abstract:** A new modification of the isolation forest called the attention-based isolation forest (ABIForest) is proposed for solving the anomaly detection problem. It incorporates an attention mechanism in the form of Nadaraya–Watson regression into the isolation forest to improve the solution of the anomaly detection problem. The main idea underlying the modification is the assignment of attention weights to each path of trees with learnable parameters depending on the instances and trees themselves. Huber's contamination model is proposed to be used to define the attention weights and their parameters. As a result, the attention weights are linearly dependent on learnable attention parameters that are trained by solving a standard linear or quadratic optimization problem. ABIForest can be viewed as the first modification of the isolation forest to incorporate an attention mechanism in a simple way without applying gradient-based algorithms. Numerical experiments with synthetic and real datasets illustrate that the results of ABIForest outperform those of other methods. The code of the proposed algorithms has been made available.

**Keywords:** anomaly detection; attention mechanism; isolation forest; Nadaraya–Watson regression; quadratic programming; contamination model

## 1. Introduction

One of the most important machine learning problems is the novelty or anomaly detection problem, which aims to detect abnormal or anomalous instances. This problem can be regarded as a challenging task because there is not a strong definition of anomalous instances, and anomalies themselves depend on certain applications. Another difficulty that defines the challenge of the problem is that anomalies usually appear only seldomly, and this fact leads to highly imbalanced training sets. Moreover, it is difficult to define a boundary between normal and anomalous observations [1]. Due to the importance of the anomaly detection problem in many applications, a huge number of papers covering anomaly detection tasks and studying various aspects of anomaly detection have been published in recent decades. Many approaches to solving the anomaly detection problem have been analyzed in comprehensive survey papers [1–11].

According to [1,12], anomalies—also referred to as abnormalities, deviants, or outliers—can be viewed as data points that are located further away from the bulk of data points, which are referred to as normal data.

The various approaches to solving the anomaly detection problem can be divided into several groups [10]. The first group consists of probabilistic and density estimation models. These include the classic density estimation models, energy-based models, and neural generative models [10]. The second largest group deals with one-class classification models. This group includes the well-known one-class classification SVMs [13–15]. The third group includes reconstruction-based models, which detect anomalies by reconstructing data instances. The well-known models from this group are autoencoders that incorrectly reconstruct anomalous instances such that the distance between an instance and its reconstruction

is larger than a predefined threshold, which is usually regarded as a hyperparameter of the model.

The next group contains distance-based anomaly detection models. One of the most popular and effective models from this group is the isolation forest (iForest) [16,17], which is a model for detecting anomalous points relative to a certain data distribution. According to iForest, anomalies are detected by using isolation which measures how far an instance is from the rest of the instances. iForest can be regarded as a tool for implementing isolation. It has linear time complexity and works well with large amounts of data. The core idea behind iForest is the tendency for anomalous instances in a dataset to be more easily separated from the rest of the sample (isolated) compared to normal instances. To isolate a data point, the algorithm recursively creates sample partitions by randomly choosing an attribute and then randomly choosing a split value for the attribute between the minimum and maximum values allowed for that attribute. The recursive partition can be represented by a tree structure called an isolation tree, while the number of partitions needed to isolate a point can be interpreted as the length of the path within the tree to the end node, starting from the root. Anomalous instances are those with a shorter path length in the tree [16,17].

iForest is a powerful tool for solving the anomaly detection problem. However, it may not take into account that some trees are built in such a way that they do not properly detect anomalous instances because of the random selection of instances for training each tree, the randomized splitting procedure, etc. It is supposed in iForest that all isolation trees behave in a similar way. However, the ranges between the trees' abilities to isolate anomalous instances may be large. Moreover, one anomalous instance can be accurately detected by a tree, whereas another anomalous instance may be undetectable by the same tree. In order to improve iForest, we propose modifying it by using an attention mechanism that can automatically distinguish the relative importance of instances and weigh it to improve the overall accuracy. The attention mechanism allows us to take different capabilities of isolation trees conditioned on instances into account by assigning attention weights to trees and instances depending on their properties. This allows us to compensate for some of the impact of "bad" trees, which are "bad" only for certain instances and can be accurate for other instances.

Attention mechanisms have been successfully applied in many applications, including natural language processing models, the computer vision area, etc. Comprehensive surveys of properties and forms of attention mechanisms and transformers can be found in [18–22].

The idea of applying an attention mechanism to iForest stems from the attention-based random forest (ABRF) models proposed in [23], where attention was implemented in the form of the Nadaraya–Watson regression [24,25] by assigning attention weights to the leaves of trees in a specific way such that the weights depended on the trees and instances. The learnable attention parameters in ABRF were trained by solving a standard quadratic optimization problem with linear constraints. It turned out that this idea of considering a random forest as Nadaraya–Watson regression [24,25] could be extended to iForest by taking the peculiarities of iForest that cause it to differ from the random forest into account. According to the original iForest, the isolation measure is estimated as the mean value of the path lengths over all trees in the forest. However, we can replace the averaging of the path lengths with Nadaraya–Watson regression, where the path length of an instance in each tree can be regarded as a prediction in the regression (the value in terms of the attention mechanism [26]), and weights (the attention weights) depend on the corresponding tree and the instance (the query in terms of the attention mechanism [26]). In other words, the final prediction of the expected path length in accordance with Nadaraya–Watson regression is a weighted sum of the path lengths over all trees. The weights of path lengths have learnable parameters (learnable attention parameters) that can be computed by minimizing a loss function of a specific form. We aim to reduce the optimization problem to a quadratic programming problem or linear programming problem for which there are many algorithms for solving. In order to achieve this aim, Huber's $\epsilon$-contamination model [27] is proposed to be used to compute

the learnable attention parameters. The contamination model allows us to represent attention weights in the form of a linear combination of the softmax operation and learnable parameters with the contamination parameter $\epsilon$, which can be viewed as probabilities. As a result, the loss function for computing learnable parameters is linear with linear constraints on the parameters as probabilities. After adding the $L_2$ regularization term, the optimization problem for computing attention weights becomes a quadratic one.

Our contributions can be summarized as follows:

1. A new modification of iForest called the attention-based isolation forest (ABIForest), which incorporates an attention mechanism in the form of the Nadaraya–Watson regression to improve the solution of the anomaly detection problem, is proposed.
2. The algorithm of computing attention weights is reduced to solving linear or quadratic programming problems due to the application of Huber's $\epsilon$-contamination model. Moreover, we propose the use of the hinge-loss function to simplify the optimization problem. The contamination parameter $\epsilon$ is regarded as a tuning hyperparameter.
3. Numerical experiments with synthetic and real datasets were performed to study ABIForest. They demonstrated outstanding results for most datasets. The code of the proposed algorithms can be found at https://github.com/AndreyAgeev/Attention-based-isolation-forest (accessed on 1 November 2022).

This paper is organized as follows. Related work can be found in Section 2. Brief introductions to the attention mechanism, Nadaraya–Watson regression, and iForest are given in Section 3. The proposed ABIForest model is considered in Section 4. Numerical experiments with synthetic and real datasets that illustrate the peculiarities of ABIForest and its comparison with iForest are provided in Section 5. Concluding remarks discussing the advantages and disadvantages of ABIForest can be found in Section 6.

## 2. Related Work

**Attention mechanism.** An attention mechanism can be viewed as an effective method for improving the performance of a large variety of machine learning models. Therefore, there are many different types of attention mechanisms depending on their applications and the models in which attention mechanisms are incorporated. The term "attention" was introduced by Bahdanau et al. [26]. Since the publication of this paper, a huge number of models based on attention mechanisms have been found in the literature. There are also several types of attention mechanisms [28], including soft and hard attention mechanisms [29], local and global attention [30], self-attention [31], multi-head attention [31], and hierarchical attention [32]. It is difficult to consider all papers devoted to attention mechanisms and their applications. Comprehensive surveys [18–22,33] have covered a large part of the available models and modifications of attention mechanisms.

Most attention models are implemented as parts of neural networks. In order to extend a set of attention models, several random forest models in which attention mechanisms were incorporated were proposed in [23,34,35]. A gradient boosting machine to which an attention mechanism was added was presented in [36].

**Anomaly detection with attention**. A wide set of machine learning tasks include anomaly detection problems. Therefore, many methods and models have been developed to address them [1–11].One of the tools for solving anomaly detection problems is an attention mechanism. Monotonic attention-based autoencoders were proposed in [37] as an unsupervised learning technique for detecting false data injection attacks. An anomaly detection method based on a Siamese network with an attention mechanism for dealing with small datasets was proposed in [38]. A so-called residual attention network that employed an attention mechanism and residual learning to improve classification efficiency and accuracy was presented in [39]. A graph anomaly detection algorithm based on attention-based deep learning for assisting the audit process was provided in [40]. Madan et al. [41] presented a novel self-supervised masked convolutional transformer block that comprised a reconstruction-based functionality. The integration of reconstruction-based functionality into a novel self-supervised predictive architectural building block was considered in [42].

Huang et al. [43] improved the efficiency and effectiveness of anomaly detection and localization during inference by using a progressive mask refinement approach that progressively uncovered the normal regions and, finally, located anomalous regions. A novel self-supervised framework for multivariate time-series anomaly detection via a graph attention network was proposed in [44]. It can be seen from the above works that the idea of applying attention in models for solving the anomaly detection problem was successfully implemented. However, attention was used in the form of components of neural networks. There are no forest-based anomaly detection models that use an attention mechanism.

**iForest.** iForest [16,17] can be viewed as one of the most important and effective methods for solving novelty and anomaly detection problems. Therefore, many modifications of the method have been developed [5] to improve it. A weighted iForest and Siamese gated recurrent unit algorithm architecture that provided a more accurate and efficient method for outlier detection of data was considered in [45]. Hariri et al. [46] proposed an extension of iForest called the extended isolation forest, which resolved issues with the assignment of anomaly scores to given data points. A theoretical framework that described the effectiveness of isolation-based approaches from a distributional viewpoint was studied in [47]. Lesouple et al. [48] presented a generalized isolation forest algorithm that generated trees without any empty branches, which significantly improved the execution times. The k-means-based iForest was developed by Karczmarek et al. [49]. This modification of iForest allowed one to build a search tree based on many branches, in contrast to the two considered in the original method. Another modification called the fuzzy set-based isolation forest was proposed in [50]. A probabilistic generalization of iForest was proposed in [51], which was based on the nonlinear dependence of a segment-cumulated probability on the length of the segment. A robust anomaly detection method called the similarity-measured isolation forest was developed by Li et al. [52] to detect abnormal segments in monitoring data. A novel hyperspectral anomaly detection method with a kernel isolation forest was proposed in [53]. The method was based on an assumption that anomalies, rather than the background, could be more susceptible to isolation in the kernel space. An improved computational framework that allows one to effectively seek the most separable attributes and spots corresponding to optimized split points was presented in [54]. Staerman et al. [55] introduced the so-called functional isolation forest, which generalized iForest to the infinite-dimensional context, i.e., the model dealt with functional random variables that took their values in a space of functions. Xu et al. [56] proposed the deep isolation forest, which was based on an isolation method with an arbitrary (linear/nonlinear) partitioning of data implemented by using neural networks.

The above works are only a some of the many extensions and modifications of iForest that have been developed due to the excellent properties of the method. However, to the best of our knowledge, there are no works considering approaches to incorporating an attention mechanism into iForest.

## 3. Preliminaries

### 3.1. Attention Mechanism as Nadaraya–Watson Regression

If we consider an attention mechanism as a method for enhancing the accuracy of iForest for the solution of the anomaly detection problem, this allows us to automatically distinguish the relative importance of features, instances, and isolation trees. According to [18,57], the original idea of attention can be understood from the statistical point of view by applying the Nadaraya–Watson kernel regression model [24,25].

Given $n$ instances $\mathcal{D} = \{(\mathbf{x}_1, y_1), \ldots, (\mathbf{x}_n, y_n)\}$, in which $\mathbf{x}_i = (x_{i1}, \ldots, x_{id}) \in \mathbb{R}^d$ is a feature vector involving $m$ features and $y_i \in \mathbb{R}$ represents the regression outputs, the task of regression is to construct a regressor $f : \mathbb{R}^m \to \mathbb{R}$ that can predict the output value $\tilde{y}$ of a new observation $\mathbf{x}$ by using available data $S$. A similar task can be formulated for classification problems.

The original idea behind the attention mechanism is to replace the simple average of outputs $\tilde{y} = n^{-1} \sum_{i=1}^{n} y_i$ for estimating the regression output $y$, corresponding to a new

input feature vector $\mathbf{x}$, with the weighted average in the form of the Nadaraya–Watson regression model [24,25]:

$$\tilde{y} = \sum_{i=1}^{n} \alpha(\mathbf{x}, \mathbf{x}_i) y_i, \tag{1}$$

where the weight $\alpha(\mathbf{x}, \mathbf{x}_i)$ conforms with the relevance of the *i*-th instance to the vector $\mathbf{x}$, i.e., it is defined in agreement with the location of the corresponding input $\mathbf{x}_i$ relative to the input variable $\mathbf{x}$ (the closer an input $\mathbf{x}_i$ is to the given variable $\mathbf{x}$, the greater $\alpha(\mathbf{x}, \mathbf{x}_i)$ will be).

In terms of the attention mechanism [26], the vectors $\mathbf{x}$, $\mathbf{x}_i$ and outputs $y_i$ are called the *query*, *keys*, and *values*, respectively. The weight $\alpha(\mathbf{x}, \mathbf{x}_i)$ is called the attention weight.

The attention weights $\alpha(\mathbf{x}, \mathbf{x}_i)$ can be defined by a normalized kernel $K$ as:

$$\alpha(\mathbf{x}, \mathbf{x}_i) = \frac{K(\mathbf{x}, \mathbf{x}_i)}{\sum_{j=1}^{n} K(\mathbf{x}, \mathbf{x}_j)}. \tag{2}$$

For a Gaussian kernel with the parameter $\omega$, the attention weights are represented through the softmax operation as:

$$\alpha(\mathbf{x}, \mathbf{x}_i) = \sigma\left(-\frac{\|\mathbf{x} - \mathbf{x}_i\|^2}{\omega}\right). \tag{3}$$

In order to enhance the attention capabilities, weights are added by trainable parameters. Several definitions of attention weights and attention mechanisms have been proposed. The most popular definitions are additive attention [26], multiplicative attention, and dot-product attention [30,31].

*3.2. Isolation Forest*

In this subsection, the main definitions of iForest are provided in accordance with the results given in [16,17]. Suppose that there is a dataset $\mathcal{D} = \{\mathbf{x}_1, \ldots, \mathbf{x}_n\}$ consisting of $n$ instances, where $\mathbf{x}_i = (x_{i1}, \ldots, x_{id}) \in \mathbb{R}^d$ is a feature vector. The isolation tree is built by using a randomly generated subset $\mathcal{D}^*$ of the dataset $\mathcal{D}$. The dataset $\mathcal{D}^*$ splits into two subsets to define a random node as follows. A feature is randomly selected by generating a random value $q$ from the set $\{1, \ldots, d\}$. Then, a split value $p$ is randomly selected from the interval $[\min_{i=1,\ldots,n} x_{iq}, \max_{i=1,\ldots,n} x_{iq}]$. With $p$ and $q$, the subset $\mathcal{D}^*$ is recursively divided into two parts at each node by using the feature number $q$ and the split value $p$—the left branch corresponds to the set with $x_{iq} \leq p$, and the right branch corresponds to the set with $x_{iq} > p$. Thus, the generated values $q$ and $p$ determine whether the data points at a node are sent down the left or the right branch. The above conditions determine the subsequent child nodes for a split node. The division stops in accordance with a rule—for example, when a branch contains a single point or when some limited depth of the tree is reached. The process of building the isolation tree begins again with a new random subsample to build another randomized tree. After building a forest consisting of $T$ trees, the training process is complete.

In the *k*-th isolation tree, an instance $\mathbf{x}$ is isolated on one of the outer nodes. A path of length $h_k(\mathbf{x})$ associated with this instance is defined as the number of nodes for which $\mathbf{x}$ goes from the root node to the leaf. Anomalous instances are those with a shorter path length in the tree. This conclusion is motivated by the fact that normal instances are more concentrated than anomalies and, thus, require more nodes to be isolated. By having the $T$ trained trees, i.e., the isolated forest, we can estimate the isolation measure as the expected path length $E[h(\mathbf{x})]$, which is computed as the mean value of the path lengths over all trees in the forest. By having the expected path length $E[h(\mathbf{x})]$, an anomaly score is defined as

$$s(\mathbf{x}, n) = 2^{-\frac{E(h(\mathbf{x}))}{c(n)}}, \tag{4}$$

where $c(n)$ is the normalizing factor, which is defined as the average value of $h(\mathbf{x})$ for a dataset of size $n$, which is computed as

$$c(n) = 2H(n-1) - \frac{2(n-1)}{n}. \tag{5}$$

Here, $H(n)$ is the $n$-th harmonic number estimated from $H(n) = \ln(n) + \delta$, where $\delta \approx 0.577216$ is the Euler–Mascheroni constant. If $n = 2$, then $c(n) = 1$.

The higher the value of $s(\mathbf{x}, n)$ is (closer to 1), the more likely it is for the instance $\mathbf{x}$ to be anomalous. If we introduce a threshold $\tau \in [0.1]$, then the condition $s(\mathbf{x}, n) > \tau$ indicates that the instance $\mathbf{x}$ is detected as an anomaly. If the condition $s(\mathbf{x}, n) \leq \tau$ is valid, then the instance $\mathbf{x}$ is likely to be normal. The threshold $\tau$ in the original iForest is taken as 0.5.

## 4. Attention-Based Isolation Forest

It should be noted that the expected path length $E[h(\mathbf{x})]$ in the original iForest is computed as the mean value of the path lengths $h_k(\mathbf{x})$ of the trees:

$$E[h(\mathbf{x})] = \frac{1}{T} \sum_{k=1}^{T} h_k(\mathbf{x}). \tag{6}$$

This method of computing the expected path length does not take into account the possible relationship between an instance and each isolation tree or the possible differences between trees. The ideas behind the attention-based RF [23] can also be applied to iForest. Therefore, our next task is to incorporate an attention mechanism into iForest.

### 4.1. Keys, Values, and Queries in iForests

First, we can point out that the outcome of each isolation tree is the path length $h_k(\mathbf{x})$, $k = 1, \ldots, n$. This implies that this outcome can be regarded as the value in the attention mechanism. Second, we define the queries and keys in iForest. Suppose that the feature vector $\mathbf{x}$ falls into the $i$-th leaf of the $k$-th tree. Let $\mathcal{J}_i^{(k)}$ be a set of indices of $n_i^{(k)}$ training instances $\mathbf{x}_j$ that also fall into the same leaf. The distance between the vector $\mathbf{x}$ and all vectors $\mathbf{x}_j$, $j \in \mathcal{J}_i^{(k)}$, shows how the vector $\mathbf{x}$ is in agreement with the corresponding vectors $\mathbf{x}_j$ and how close it is to the vectors $\mathbf{x}_j$ from the same leaf. If the distance is small, then we can conclude that the vector $\mathbf{x}$ is well performed by the $k$-th tree. The distance between the vector $\mathbf{x}$ and all vectors $\mathbf{x}_j$, $j \in \mathcal{J}_i^{(k)}$, can be represented as a distance between the vector $\mathbf{x}$ and the mean values of all vectors $\mathbf{x}_j$ with indices $j \in \mathcal{J}_i^{(k)}$. The mean vector of $\mathbf{x}_j$ with indices $j \in \mathcal{J}_i^{(k)}$ can be viewed as a characteristic of the corresponding path, i.e., this vector characterizes a group of instances that fall into the corresponding leaf. Hence, the mean vector shows how the vector $\mathbf{x}$ is in agreement with this group. If we denote the mean value of $\mathbf{x}_j$, $j \in \mathcal{J}_i^{(k)}$ as $\mathbf{A}_k(\mathbf{x})$, then it holds that

$$\mathbf{A}_k(\mathbf{x}) = \frac{1}{n_i^{(k)}} \sum_{j \in \mathcal{J}_i^{(k)}} \mathbf{x}_j. \tag{7}$$

We omit the index $j$ in $\mathbf{A}_k(\mathbf{x})$ because the instance $\mathbf{x}$ can fall only into one leaf of each tree.

The vectors $\mathbf{A}_k(\mathbf{x})$ and $\mathbf{x}$ can be regarded as the key and the query, respectively. Then, (6) can be rewritten by using the attention weights $\alpha(\mathbf{x}, \mathbf{A}_k(\mathbf{x}), \mathbf{w})$ as follows:

$$E[h(\mathbf{x})] = \sum_{k=1}^{T} \alpha(\mathbf{x}, \mathbf{A}_k(\mathbf{x}), \mathbf{w}) \cdot h_k(\mathbf{x}), \tag{8}$$

where $\alpha(\mathbf{x}, \mathbf{A}_k(\mathbf{x}), \mathbf{w})$ conforms with the relevance of the "mean instance" $\mathbf{A}_k(\mathbf{x})$ for the vector $\mathbf{x}$ and satisfies the following condition:

$$\sum_{k=1}^{T} \alpha(\mathbf{x}, \mathbf{A}_k(\mathbf{x}), \mathbf{w}) = 1, \ \alpha(\mathbf{x}, \mathbf{A}_k(\mathbf{x}), \mathbf{w}) \geq 0, \ k = 1, \dots, T. \tag{9}$$

We replaced the expected path length (6) with the weighted sum of path lengths (8) such that the weights $\alpha$ depend on $\mathbf{x}$, the mean vector $\mathbf{A}_k(\mathbf{x})$, and the vector of parameters $\mathbf{w}$. The vector $\mathbf{w}$ in the attention weights represents the trainable attention parameters. Their values depend on the dataset and on the isolation tree's properties. If we return to the Nadaraya–Watson kernel regression model, then the expected path length $E[h(\mathbf{x})]$ can be viewed as the regression output, and the path lengths $h_k(\mathbf{x})$ of all trees for query $\mathbf{x}$ are predictions (values in terms of the attention mechanism [26]).

Suppose that the trainable parameters $\mathbf{w}$ belong to a set $\mathcal{W}$. Then, they can be found by solving the following optimization problem:

$$\mathbf{w}_{opt} = \arg \min_{\mathbf{w} \in \mathcal{W}} \ \sum_{s=1}^{n} L\big(E[h(\mathbf{x}_s)], \mathbf{x}_s, \mathbf{w}\big). \tag{10}$$

Here, $L\big(E[h(\mathbf{x}_s)], \mathbf{x}_s, \mathbf{w}\big)$ is the loss function, whose definition, as well as the definition of $\alpha(\mathbf{x}, \mathbf{A}_k(\mathbf{x}), \mathbf{w})$, is the next task.

A general scheme of ABIForest is shown in Figure 1. The red branches show the paths of an anomalous instance $\mathbf{x}$ that falls into the corresponding red leaves. For these leaves, the keys $\mathbf{A}_1(\mathbf{x}), \dots, \mathbf{A}_T(\mathbf{x})$, values $h_1(\mathbf{x}), \dots, h_T(\mathbf{x})$, and attention weights $\alpha(\mathbf{x}, \mathbf{A}_k(\mathbf{x}), \mathbf{w})$, $k = 1, \dots, T$, are computed. The products $\alpha(\mathbf{x}, \mathbf{A}_k(\mathbf{x}), \mathbf{w}) \cdot h_k(\mathbf{x})$ are summed in accordance with Nadaraya–Watson regression to obtain the expected path length $E[h(\mathbf{x})]$, which is compared with the threshold $\gamma$ to make a decision on whether instance $\mathbf{x}$ is normal or anomalous.

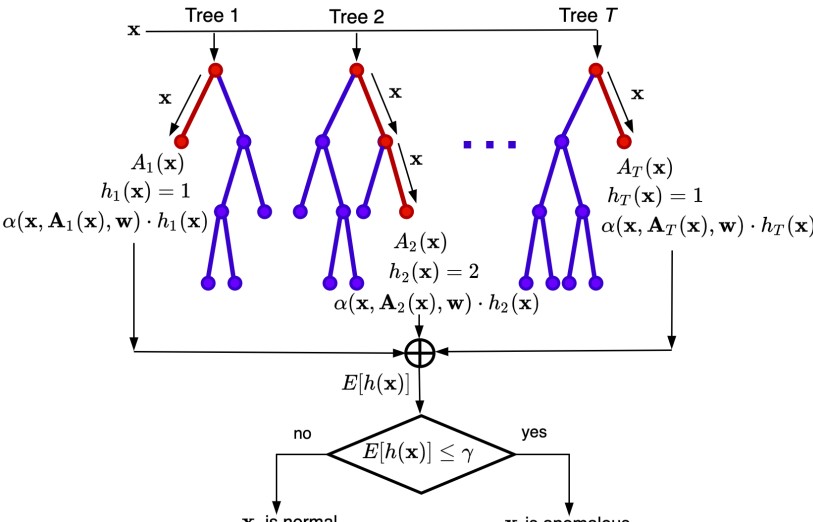

**Figure 1.** A general scheme of ABIForest, which illustrates how iForest is modified by incorporating an attention mechanism.

### 4.2. Loss Function and Attention Weights

First, we reformulate the decision rule ($s(\mathbf{x}, n) > \tau$) for determining anomalous instances by establishing a similar condition for $E[h(\mathbf{x})]$. Suppose that $\gamma$ is a threshold such

that the condition $E[h(\mathbf{x})] \leq \gamma$ indicates that instance $\mathbf{x}$ is detected as an anomaly. Then, it follows from (4) that $\gamma$ can be expressed through the threshold $\tau$ as:

$$\gamma = -c(n) \cdot \log_2(\tau). \tag{11}$$

Hence, we can write the decision rule about the anomaly as follows:

$$\text{decision} = \begin{cases} \text{anomalous,} & \text{if } E[h(\mathbf{x})] - \gamma \leq 0, \\ \text{normal,} & \text{otherwise.} \end{cases} \tag{12}$$

We also introduce the instance label $y_s$, which is 1 if the training instance $\mathbf{x}_s$ is anomalous and $-1$ if it is normal. If labels are not known, then prior values of labels can be determined by using the original iForest.

We propose the following loss function:

$$L(h(\mathbf{x}_s), \mathbf{x}_s, \mathbf{w}) = \max(0, y_s(E[h(\mathbf{x}_s)] - \gamma)). \tag{13}$$

It can be seen from (13) that the loss function is 0 if $E[h(\mathbf{x}_s)] - \gamma$ and $y_s$ have different signs, i.e., if the decision about an anomalous (normal) instance coincides with the corresponding label. Substituting (8) into (13), we rewrite the optimization problem (10) as:

$$\mathbf{w}_{opt} = \arg\min_{\mathbf{w} \in \mathcal{W}} \left[ \sum_{s=1}^{n} \max\left( 0, y_s \left( \sum_{k=1}^{T} \alpha(\mathbf{x}, \mathbf{A}_k(\mathbf{x}), \mathbf{w}) \cdot h_k(\mathbf{x}) - \gamma \right) \right) \right]. \tag{14}$$

An important question is that of how to simplify the above problem to get a unique solution and how to define the attention weights $\alpha(\mathbf{x}, \mathbf{A}_k(\mathbf{x}), \mathbf{w})$ depending on the trainable parameters $\mathbf{w}$. This can be done by using Huber's $\epsilon$-contamination model.

### 4.3. Huber's Contamination Model

We propose the use of a simple representation of attention weights presented in [23], which is based on applying Huber's $\epsilon$-contamination model [27]. The model is represented as a set of discrete probability distributions $F$ of the form:

$$F = (1 - \epsilon) \cdot P + \epsilon \cdot R, \tag{15}$$

where $P = (p_1, \ldots, p_T)$ is a discrete probability distribution contaminated by another probability distribution, which is denoted as $R = (r_1, \ldots, r_T)$, under condition that the probability distribution $R$ can be arbitrary; the contamination parameter $\epsilon \in [0, 1]$ controls the degree of the contamination.

The contaminating distribution $R$ is a point in the unit simplex with $T$ vertices, which are denoted as $S(1, T)$. The distribution $F$ is a point in a small simplex that belongs to the unit simplex. The size of the small simplex depends on the hyperparameter $\epsilon$. If $\epsilon = 1$, then the small simplex coincides with the unit simplex. If $\epsilon = 0$, then the small simplex is reduced to a single distribution $P$.

We propose the consideration of every element of $P$ as a result of the softmax operation

$$p_k = \sigma\left( -\frac{\|\mathbf{x} - \mathbf{A}_k(\mathbf{x})\|^2}{\omega} \right), \tag{16}$$

that is,

$$P = \left( \sigma\left( -\frac{\|\mathbf{x} - \mathbf{A}_1(\mathbf{x})\|^2}{\omega} \right), \ldots, \sigma\left( -\frac{\|\mathbf{x} - \mathbf{A}_T(\mathbf{x})\|^2}{\omega} \right) \right).$$

Moreover, we propose the consideration of the distribution $R$ as the vector of trainable parameters $\mathbf{w}$, that is,

$$R = \mathbf{w} = (w_1, \ldots, w_T).$$

Hence, the attention weight $\alpha(\mathbf{x}, \mathbf{A}_k(\mathbf{x}), \mathbf{w})$ can be represented for every $k = 1, \ldots, T$ as follows:

$$\alpha(\mathbf{x}, \mathbf{A}_k(\mathbf{x}), \mathbf{w}) = (1 - \epsilon) \cdot \sigma\left(-\frac{\|\mathbf{x} - \mathbf{A}_k(\mathbf{x})\|^2}{\omega}\right) + \epsilon \cdot w_k. \tag{17}$$

An important property of the above representation is that the attention weight linearly depends on the trainable parameters, and the softmax operation depends only on the hyperparameter $\omega$. The trainable parameters $\mathbf{w} = (w_1, \ldots, w_T)$ are restricted by the unit simplex $S(1, T)$ and, therefore, $\mathcal{W} = S(1, T)$. This implies that the constraints for $\mathbf{w}$ are linear ($w_i \geq 0$ and $w_1 + \cdots + w_T = 1$).

### 4.4. Loss Function with the Contamination Model

Let us substitute the obtained expression (17) for the attention weight $\alpha(\mathbf{x}, \mathbf{A}_k(\mathbf{x}), \mathbf{w})$ into the objective function (14). After simplification, we get

$$\min_{\mathbf{w} \in S(1,T)} \sum_{s=1}^{n} \max\left(0, D_s(\epsilon, \omega) + y_s \epsilon \sum_{k=1}^{T} h_k(\mathbf{x}_s) w_k\right) \tag{18}$$

where

$$D_s(\epsilon, \omega) = y_s(1 - \epsilon) \sum_{k=1}^{T} \sigma\left(-\frac{\|\mathbf{x}_s - \mathbf{A}_k(\mathbf{x}_s)\|^2}{\omega}\right) - \gamma T. \tag{19}$$

Let us introduce the new variables

$$v_s = \max\left(0, D_s(\epsilon, \omega) + y_s \epsilon \sum_{k=1}^{T} h_k(\mathbf{x}_s) w_k\right). \tag{20}$$

Then, the problem (18) can be rewritten as follows:

$$\min \sum_{s=1}^{n} v_s, \tag{21}$$

subject to

$$v_s \geq D_s(\epsilon, \omega) + y_s \epsilon \sum_{k=1}^{T} h_k(\mathbf{x}_s) w_k, \tag{22}$$

$$v_s \geq 0, \ s = 1, \ldots, n, \tag{23}$$

$$w_1 + \cdots + w_T = 1, \ w_k \geq 0, \ k = 1, \ldots, T. \tag{24}$$

This is a linear optimization problem with the optimization variables $w_1, \ldots, w_T$ and $v_1, \ldots, v_n$.

The optimization problem can be improved by adding a regularization term $\|\mathbf{w}\|^2$ with the hyperparameter $\lambda$, which controls the strength of the regularization. In this case, the optimization problem becomes

$$\min \sum_{s=1}^{n} v_s + \lambda \|\mathbf{w}\|^2, \tag{25}$$

subject to (22)–(24).

We obtain a standard quadratic programming problem whose solution does not meet any difficulties.

## 5. Numerical Experiments

The proposed attention-based iForest was studied by using synthetic and real data and was compared with the original iForest. A brief introduction to these datasets is given

in Table 1, where $d$ is the number of features and $n_{norm}$ and $n_{anom}$ are numbers of normal and anomalous instances, respectively.

Different values for hyperparameters, including the threshold $\tau$, the number of trees in the forest, the contamination parameter $\epsilon$, and the kernel parameter $\omega$, were tested, and those leading to the best results were chosen. In particular, the hyperparameter $\epsilon$ in ABIForest took values of 0, 0.25, 0.5, 0.75, and 1; the hyperparameter $\gamma$ changed from 0.5 to 0.7; the hyperparameter $\omega$ took values of 0.1, 10, 20, 30, and 40. The F1-score was used as a measure of the anomaly detection accuracy. It was used because the number of anomalous instances was significantly less than the number of normal instances, i.e., because the normal and anomalous instances were imbalanced. To evaluate the F1-score, a cross-validation with 100 repetitions was performed, where in each run, 66.7% of the data were randomly selected for training ($2n/3$) and 33.3% were randomly selected for testing ($n/3$). The numerical results are presented in the tables, and the best results are shown in bold.

**Table 1.** A brief introduction to the datasets.

| Dataset | $n_{norm}$ | $n_{anom}$ | $d$ |
|---|---|---|---|
| Circle (synthetic) | 1000 | 200 | 2 |
| Normal dataset (synthetic) | 1000 | 50 | 2 |
| Credit | 1500 | 400 | 30 |
| Ionosphere | 225 | 126 | 33 |
| Arrhythmia | 386 | 66 | 18 |
| Mulcross | 1800 | 400 | 4 |
| Http | 500 | 50 | 3 |
| Pima | 500 | 268 | 8 |

### 5.1. Synthetic Datasets

The first synthetic dataset used for the numerical experiments was the Circle dataset. Its points were divided into two parts concentrated around small and large circles, as shown in Figure 2, where the training and testing sets are depicted in the left and right pictures, respectively. In order to optimize the model parameters in the numerical experiments, we performed cross-validation. Gaussian noise with a standard deviation of 0.1 was added to the data for all experiments.

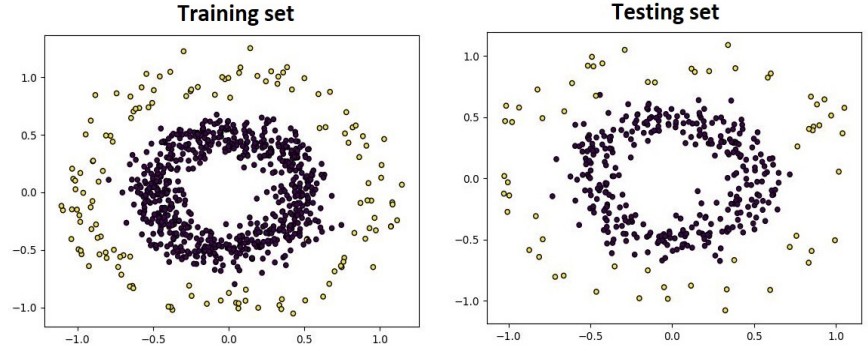

**Figure 2.** Points from the Circle dataset.

The second synthetic dataset (the Normal dataset) contained points generated from normal distributions with two expectations: $(-2, -2)$ and $(2, 2)$. Anomalies were generated

from a uniform distribution in the interval $[-1, 1]$. The training and testing sets are depicted in Figure 3.

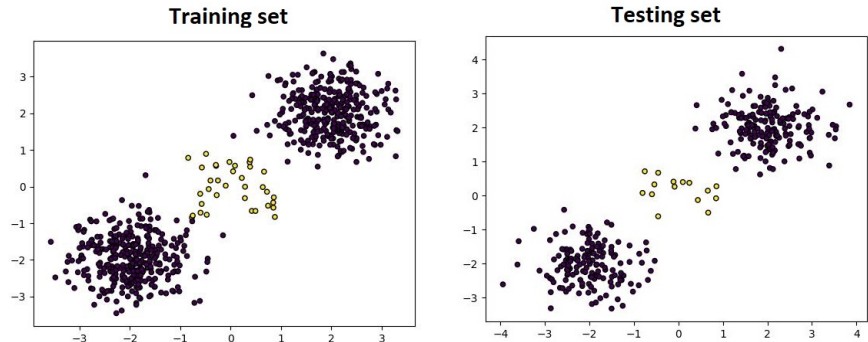

**Figure 3.** Points from the Normal dataset.

First, we studied the Circle dataset. The F1-score measures obtained for ABIForest are shown in Table 2, where the F1-score is presented as a function of the hyperparameters $\epsilon$ and $\tau$ with the number of trees in the isolation forest of $T = 150$. It is interesting to note that ABIForest was sensitive to changes in $\tau$, whereas $\epsilon$ did not significantly impact the results. For comparison purposes, the F1-score measures of the original iForest as a function of the number of trees $T$ and the hyperparameter $\tau$ are shown in Table 3. It can be seen from Table 3 that the largest value of the F1-score was achieved with 150 trees in the forest and with $\tau = 0.5$. One can also see from Tables 2 and 3 that ABIForest provided results that outperformed the same results of the original iForest.

**Table 2.** F1-scores of ABIForest consisting of $T = 150$ trees as a function of the hyperparameters $\tau$ and $\epsilon$ for the Circle dataset with $\omega = 20$.

| $\epsilon$ | | $\tau$ | |
| --- | --- | --- | --- |
| | 0.5 | 0.6 | 0.7 |
| 0.0 | 0.276 | 0.973 | 0.236 |
| 0.25 | 0.2749 | 0.975 | 0.162 |
| 0.5 | 0.273 | **0.978** | 0.100 |
| 0.75 | 0.273 | 0.975 | 0.062 |
| 1.0 | 0.271 | 0.973 | 0.037 |

**Table 3.** F1-scores of the original iForest as a function of the number of trees $T$ and the hyperparameter $\tau$ for the Circle dataset.

| $\tau$ | | | $T$ | | |
| --- | --- | --- | --- | --- | --- |
| | 5 | 15 | 25 | 50 | 150 |
| 0.3 | 0.270 | 0.270 | 0.270 | 0.270 | 0.270 |
| 0.4 | 0.286 | 0.273 | 0.271 | 0.270 | 0.270 |
| 0.5 | 0.729 | 0.864 | 0.899 | 0.906 | **0.920** |
| 0.6 | 0.639 | 0.603 | 0.598 | 0.603 | 0.606 |

Similar numerical experiments with the Normal dataset are presented in Tables 4 and 5. We can see again that ABIForest outperformed iForest, namely, the best value of the F1-score

provided by iForest was 0.252, whereas the best value of the F1-score for ABIForest was 0.413, and this result was obtained with $\omega = 20$.

**Table 4.** F1-scores of ABIForest consisting of $T = 150$ trees as a function of the hyperparameters $\tau$ and $\epsilon$ for the Normal dataset with $\omega = 20$.

| $\epsilon$ | $\tau$ | | |
|---|---|---|---|
| | 0.5 | 0.6 | 0.7 |
| 0.0 | 0.099 | 0.410 | 0.0 |
| 0.25 | 0.147 | 0.410 | 0.162 |
| 0.5 | 0.177 | **0.413** | 0.0 |
| 0.75 | 0.176 | 0.412 | 0.0 |
| 1.0 | 0.178 | 0.408 | 0.0 |

**Table 5.** F1-scores of the original iForest as a function of the number of trees $T$ and the hyperparameter $\tau$ for the Normal dataset.

| $\tau$ | $T$ | | | | |
|---|---|---|---|---|---|
| | 5 | 15 | 25 | 50 | 150 |
| 0.3 | 0.082 | 0.082 | 0.082 | 0.082 | 0.082 |
| 0.4 | 0.088 | 0.083 | 0.083 | 0.082 | 0.082 |
| 0.5 | 0.220 | 0.248 | 0.249 | 0.250 | **0.252** |
| 0.6 | 0.191 | 0.141 | 0.091 | 0.040 | 0.021 |

Figure 4 illustrates how the F1-score depended on the hyperparameter $\tau$ for the Circle dataset. The corresponding functions are depicted for different contamination parameters $\epsilon$, and they were obtained for the case of $T = 150$ trees in iForest. It can be seen from Figure 4 that the largest value of the F1-score was achieved with $\omega = 20$ and $\epsilon = 0.5$. It can also be seen from the results in Figure 4 that the F1-score significantly depended on the hyperparameter $\omega$, especially for small values of $\epsilon$. The F1-score measures as functions of the contamination parameter $\epsilon$ for different numbers of trees in iForest $T$ for the Circle dataset obtained with the hyperparameters $\gamma = 0.6$ and $\omega = 20$ are depicted in Figure 5.

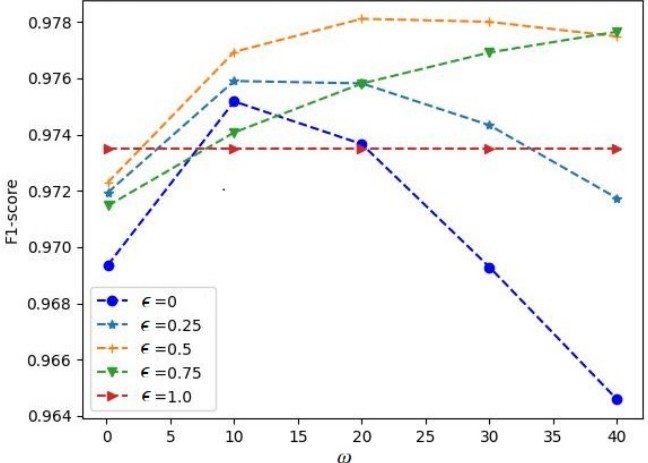

**Figure 4.** F1-score measures as functions of the softmax hyperparameter $\omega$ for different contamination parameters $\epsilon$ for the Circle dataset.

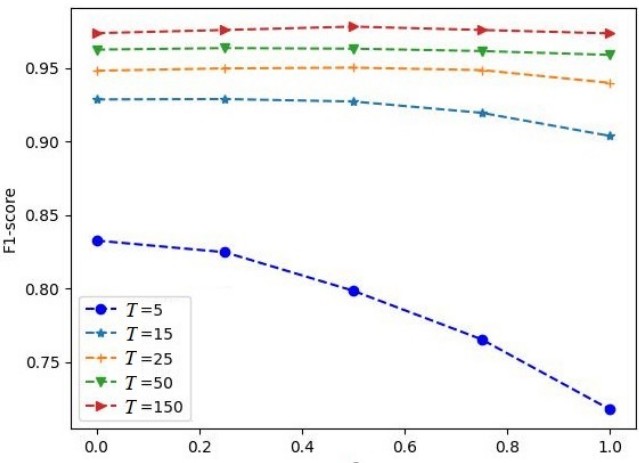

**Figure 5.** F1-score measures as functions of the contamination parameter $\epsilon$ for different numbers of trees in iForest $T$ for the Circle dataset.

Figure 6 illustrates the results of a comparison between iForest and ABIForest on the basis of the test set, which is depicted in the left panel of Figure 6. The predictions obtained by iForest consisting of 150 trees with $\tau = 0.5$ are depicted in the central panel. The predictions obtained by ABIForest with $\epsilon = 0.5$, $\tau = 0.6$, and $\omega = 0.1$ are shown in the right panel. One can see in Figure 6 that some points in the central panel were incorrectly identified as anomalous ones, whereas ABIForest correctly classified them as normal instances. Figure 6 should not be considered as a single realization that defines the F1-score. It is one of many cases corresponding to different generations of test sets; therefore, the numbers of normal and anomalous instances can be different in each realization.

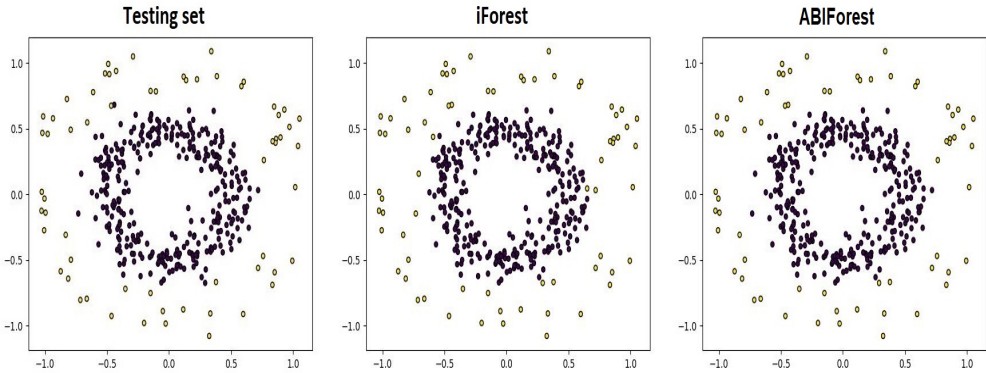

**Figure 6.** Comparison of the test set generated for the Circle dataset (the **left** panel), predictions obtained by iForest (the **central** panel), and predictions obtained by ABIForest (the **right** panel).

Similar dependencies for the Normal dataset are shown in Figures 7 and 8. However, it follows from Figure 7 that the largest values of the F1-score were achieved for $\epsilon = 0$. This implies that the main contribution to the attention weights was caused by the softmax operation. The F1-score measures shown in Figure 8 were obtained with the hyperparameters $\gamma = 0.6$ and $\omega = 20$.

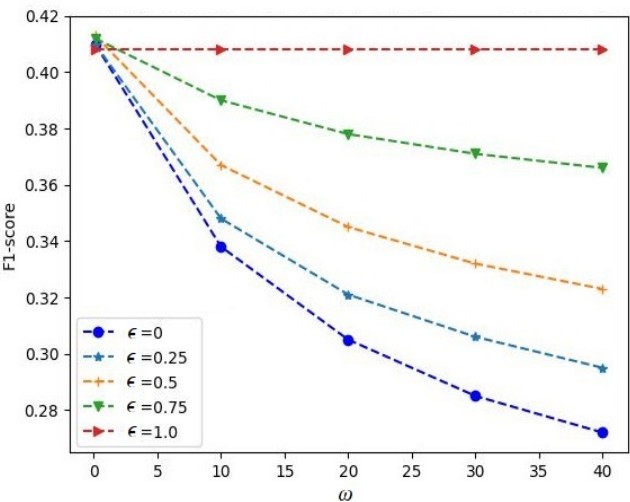

**Figure 7.** F1-score measures as functions of the softmax hyperparameter $\omega$ for different contamination parameters $\epsilon$ and for the Normal dataset.

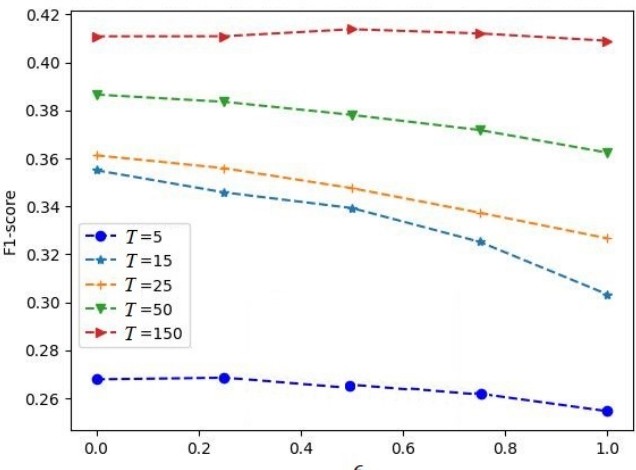

**Figure 8.** F1-score measures as functions of the contamination parameter $\epsilon$ for different numbers of trees in iForest $T$ for the Circle dataset.

The results of a comparison between iForest and ABIForest for the Normal dataset are shown in Figure 9, where a realization of the test set and the predictions of iForest and ABIForest are shown in the left, central, and right panels, respectively. The predictions were obtained by means of iForest consisting of 150 trees with $\tau = 0.5$ and by ABIForest consisting of the same number of trees with $\epsilon = 0.5$, $\tau = 0.6$, and $\omega = 0.1$.

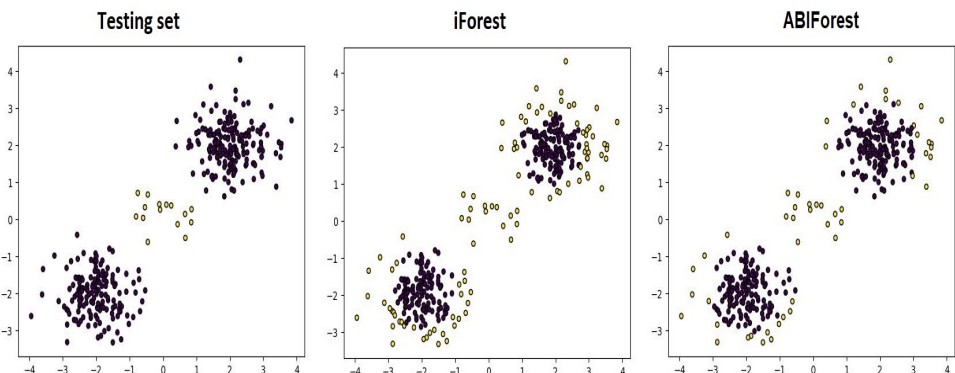

**Figure 9.** Comparison of the test set generated for the Normal dataset (the **left** panel), predictions obtained by iForest (the **central** panel), and predictions obtained by ABIForest (the **right** panel).

Another interesting question is that of how the prediction accuracy of ABIForest depends on the size of the training data. The corresponding results for the synthetic datasets are shown in Figure 10, where the solid and dashed lines correspond to the F1-scores of iForest and ABIForest, respectively. The numbers of trees in all experiments were taken as $T = 150$. The same results are also given in numerical form in Table 6. It can be seen in Figure 10 for the Circle dataset that the F1-score of iForest decreased with the increase in the number of training data after $n = 200$. This was because the number of trees ($T = 150$) was fixed, and the trees could not be improved. This effect was discussed in [17], where the problems of swamping and masking were studied. The authors of [17] considered subsampling to overcome these problems. One can see in Figure 10 that ABIForest coped with this difficulty. Another behavior of ABIForest could be observed for the Normal dataset, which was characterized by two clusters of normal points. In this case, the F1-score decreased as $n$ increased, and then increased with $n$.

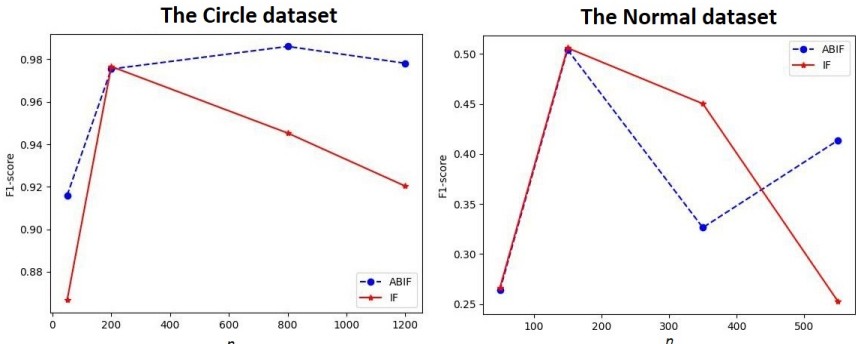

**Figure 10.** Illustration how the F1-score measures of iForest and ABIForest depend on the number of training data for the Circle dataset (the **left** panel) and the Normal dataset (the **right** panel).

**Table 6.** F1-score measures of the original iForest and ABIForest as functions of the training data number $n$ for the Circle and Normal datasets.

| The Circle Dataset | | | | |
|---|---|---|---|---|
| $n$ | 50 | 200 | 800 | 1200 |
| iForest | 0.867 | **0.977** | 0.945 | 0.920 |
| ABIForest | 0.916 | 0.975 | **0.986** | 0.978 |
| The Normal dataset | | | | |
| $n$ | 50 | 150 | 350 | 550 |
| iForest | 0.266 | **0.506** | 0.450 | 0.253 |
| ABIForest | 0.264 | **0.512** | 0.326 | 0.413 |

*5.2. Real Datasets*

The first real dataset that was used in the numerical experiments is called the Credit dataset (https://www.kaggle.com/datasets/mlg-ulb/creditcardfraud, (accessed on 1 November 2022). According to the dataset's description, it contains transactions made by credit cards in September 2013 by European cardholders, with 492 fraud cases out of 284,807 transactions. We used only 1500 normal instances and 400 anomalous ones, which were randomly selected from the whole Credit dataset. The second dataset, called the Ionosphere dataset (https://www.kaggle.com/datasets/prashant111/ionosphere, (accessed on 1 November 2022)) is a collection of radar returns from the ionosphere. The next dataset is called the Arrhythmia dataset (https://www.kaggle.com/code/medahmedkrichen/arrhythmia-classification, (accessed on 1 November 2022)).The smallest classes with numbers of 3, 4, 5, 7, 8, 9, 14, and 15 were combined to form outliers in the Arrhythmia dataset. The Mulcross dataset (https://github.com/dple/Datasets, (accessed on 1 November 2022)) was generated from a multivariate normal distribution with two dense anomaly clusters. We used 1800 normal and 400 anomalous instances. The Http dataset (https://github.com/dple/Datasets, (accessed on 1 November 2022)) was used in [17] to study iForest. The Pima dataset (https://github.com/dple/Datasets, (accessed on 1 November 2022)) aims to predict whether or not a patient has diabetes. The Credit, Mulcross, and Http datasets were reduced to simplify the experiments.

The numerical results are shown in Table 7. ABIForest is presented in Table 7 with the hyperparameters $\epsilon$, $\tau$, and $\omega$, as well as the F1-score. iForest is presented with the hyperparameter $\tau$ and the corresponding F1-score. The hyperparameters leading to the largest F1-score are presented in Table 7. It can be seen from Table 7 that ABIForest provided outstanding results for five of the six datasets. It is also interesting to point out that the optimal values of the hyperparameter $\epsilon$ for the Ionosphere and Mullcross datasets were equal to 0. This implies that the attention weights were entirely determined by the softmax operation (see (17)). A contrary case was when $\epsilon_{opt} = 1$. In this case, the softmax operations and their parameter $\omega$ were not used, and the attention weights were entirely determined by the parameters **w**, which could be regarded as the weights of trees.

**Table 7.** F1-score measures of ABIForest consisting of $T = 150$ trees for different real datasets with the optimal values of $\tau$, $\epsilon$, and $\omega$ and the F1-score measures of iForest with the optimal values of $\tau$.

| Dataset | ABIForest | | | | iForest | |
| --- | --- | --- | --- | --- | --- | --- |
| | $\epsilon_{opt}$ | $\tau_{opt}$ | $\omega_{opt}$ | F1 | $\tau_{opt}$ | F1 |
| Credit | 0.25 | 0.55 | 0.1 | **0.911** | 0.4 | 0.836 |
| Ionosphere | 0.0 | 0.4 | 0.1 | **0.693** | 0.45 | 0.684 |
| Arrhythmia | 1.0 | 0.45 | — | **0.481** | 0.4 | 0.479 |
| Mullcross | 0.0 | 0.6 | 0.1 | 0.507 | 0.5 | **0.516** |
| Http | 0.75 | 0.55 | 0.1 | **0.843** | 0.5 | 0.720 |
| Pima | 0.75 | 0.45 | 30 | **0.553** | 0.4 | 0.540 |

It is interesting to study how the hyperparameter $\tau$ impacts the performance of ABIForest and iForest. The corresponding dependencies are depicted in Figures 11–13. The results of comparisons were obtained under the conditions of the optimal values of $\epsilon$ and $\omega$ given in Table 7. One can see in Figure 11 that $\tau$ differently impacted the performance of ABIForest and iForest for the Credit dataset, whereas the corresponding dependencies scarcely differed for the Ionosphere dataset. This peculiarity was caused by the optimal values of the contamination parameter $\epsilon$. It can be seen from Table 7 that $\tau_{opt} = 0$ for the Ionosphere dataset. This implies that the attention weights were determined only by the softmax operations, which weakly impacted the model performance, and their values were close to $1/T$. Moreover, the Ionosphere dataset was one of the smallest datasets, with a large number of anomalous instances (see Table 1). Therefore, additional learnable parameters may lead to overfitting. This is a reason for why the optimal hyperparameter $\epsilon$ did not impact the model performance. It is also interesting to note that the optimal value of the contamination parameter for the Mullcross dataset was 0 (see Table 7). However, one can see quite different dependencies in the right panel of Figure 12. This was caused by the large impact of the softmax operations, whose values were far from $1/T$, and they provided results that were different from those of iForest.

Generally, one can see in Figures 11–13 that the models strongly depended on the hyperparameters $\tau$ and $\epsilon$. Most of the dependencies illustrated that there was an optimal value of $\tau$ for each case that was close to 0.5 for iForest and for ABIForest. The same can be said about the contamination parameter $\epsilon$.

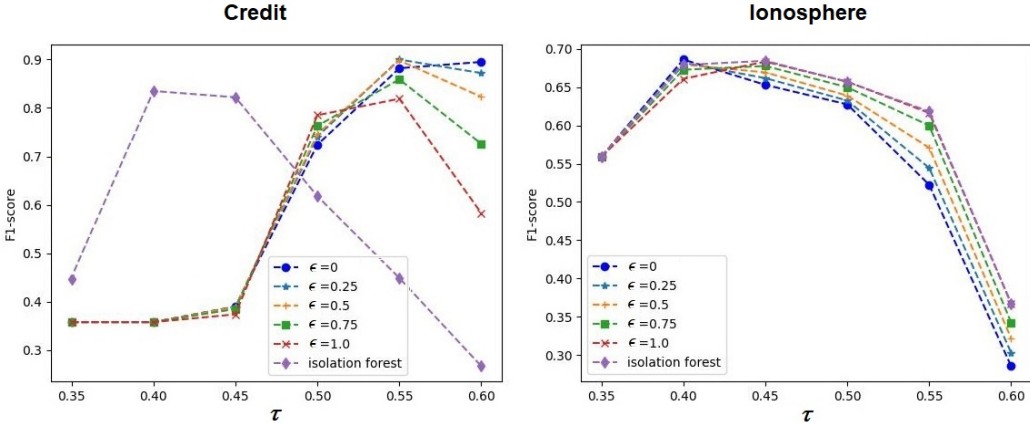

**Figure 11.** Comparison of iForest and ABIForest with different thresholds $\tau$ and with different contamination parameters $\epsilon$ for the Credit (the **left** panel) and Ionosphere (the **right** panel) datasets.

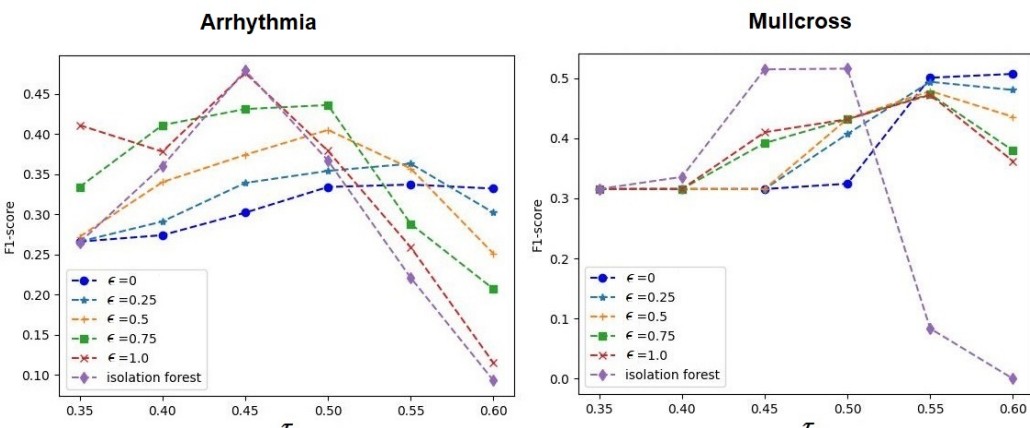

**Figure 12.** Comparison of iForest and ABIForest with different thresholds $\tau$ and with different contamination parameters $\epsilon$ for the Arrhythmia (the **left** panel) and Mullcross (the **right** panel) datasets.

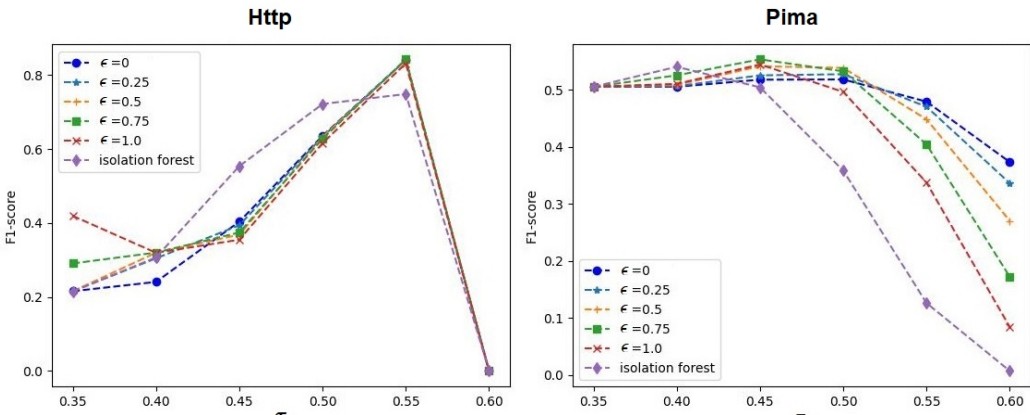

**Figure 13.** Comparison of iForest and ABIForest with different thresholds $\tau$ and with different contamination parameters $\epsilon$ for the Http (the **left** panel) and Pima (the **right** panel) datasets.

In order to study how ABIForest performed in comparison with iForest by changing the number of instances, we considered the Ionosphere dataset. Table 8 shows the F1-score measures obtained by iForest and ABIForest for different numbers of instances that were randomly selected from the dataset under the condition that the ratio of normal and anomalous instances was not changed. It can be seen from Table 8 that ABIForest provided better results in comparison with those of iForest. Moreover, the difference between the F1-score measures increased as $n$ decreased. It is also interesting to note that iForest outperformed ABIForest by $n = 80$. This can be explained by the overfitting of ABIForest due to the training parameters (vector **w**). Another question is that of how the number of anomalous instances impacts the performance of ABIForest under the condition that the number of normal instances is not changed. The corresponding numerical results are shown in Table 9. It is important to point out that ABIForest outperformed iForest for all numbers of anomalous instances. Moreover, the difference between the F1-score measures for ABIForest and iForest increased as $n_{anom}$ decreased.

**Table 8.** F1-score measures of the original iForest and ABIForest as functions of the number of instances $n$ in the Ionosphere dataset under the condition that the ratio of normal and anomalous instances was not changed.

| The Ionosphere Dataset | | | | |
|---|---|---|---|---|
| $n$ | 80 | 100 | 200 | 300 |
| iForest | 0.488 | 0.492 | 0.588 | 0.677 |
| ABIForest | 0.459 | 0.602 | 0.674 | 0.690 |

**Table 9.** F1-score measures of the original iForest and ABIForest as functions of the number of anomalous instances $n_{anom}$ in the Ionosphere dataset under the condition that the number $n_{norm}$ of normal instances was not changed.

| The Ionosphere Dataset | | | | | |
|---|---|---|---|---|---|
| $n_{anom}$ | 10 | 20 | 40 | 50 | 60 |
| iForest | 0.655 | 0.690 | 0.694 | 0.687 | 0.681 |
| ABIForest | 0.692 | 0.709 | 0.711 | 0.695 | 0.687 |

## 6. Concluding Remarks

A new modification of iForest by using an attention mechanism has been proposed. Let us focus on the advantages and disadvantages of the modification.

**Advantages:**

1. ABIForest is very simple from the point of view of computation because, in contrast to an attention-based neural network, the attention weights in ABIForest are trained by solving a standard quadratic optimization problem. This modification avoids the use of gradient-based algorithms to compute the optimal learnable attention parameters.
2. ABIForest is a flexible model that can be simply modified. There are several components of ABIForest that can be changed to improve the model's performance. First, different kernels can be used instead of the Gaussian kernel considered above. Second, there are statistical models [58] that are different from Huber's $\epsilon$-contamination model that can also be used in ABIForest. Third, the attention weights can be associated with some subsets of trees, including intersecting subsets. In this case, the number of trainable parameters can be reduced to avoid overfitting. Fourth, the paths in trees can be also attended, for example, by assigning attention weights to each branch in every path. Fifth, multi-head attention can be applied to iForest in order to improve the model—for example, by changing the hyperparameter $\omega$ of the softmax. Sixth, the distance between the instance **x** and all instances that fall in the same leaf as **x** can be differently defined. The above improvements can be regarded as directions for further research.
3. The attention model is trained after building the forest. This implies that we do not need to rebuild iForest to achieve higher accuracy. The hyperparameters are tuned without rebuilding iForest. Moreover, we can apply various modifications and extensions of iForest and incorporate the attention mechanism in the same way as in the original iForest.
4. ABIForest allows us to obtain an interpretation that answers the question of why an instance is anomalous. This can be done by analyzing the isolation trees with the largest attention weights.
5. ABIForest deals perfectly with tabular data.
6. It follows from the numerical experiments that ABIForest improves the performance of iForest for many datasets.

**Disadvantages:**

1. The main disadvantage is that ABIForest has three additional hyperparameters: the contamination parameter $\epsilon$, the hyperparameter of the softmax operation $\omega$, and the regularization hyperparameter $\lambda$. We do not include the threshold $\tau$, which is also used in iForest. Additional hyperparameters lead to significant increases in the validation time.
2. Some additional time is required to solve the optimization problem (14).
3. In contrast to iForest, ABIForest is a supervised model. It requires one to have labels of data (normal or anomalous) in order to determine the criteria of optimization, that is, to construct the optimization problem (14).

It is important to point out that small improvements were observed for many real datasets. However, with these improvements, ABIForest was able to detect additional anomalous instances that were not detected by the original iForest. Moreover, we applied the simplest form of an attention mechanism that used a contamination model to simplify computations. However, when anomalous instances are crucial for a certain application, the attention mechanism can be significantly improved by introducing additional learnable parameters that make the whole algorithm more accurate. The proposed simple attention-based model can be regarded as a first step in incorporating various forms of attention mechanisms into iForest. This aims to illustrate that this incorporation can be a prospective tool in the anomaly detection area of research. In spite of the disadvantages considered above, ABIForest can be viewed as the first version for incorporating an attention mechanism into iForest, which showed results that outperformed those of the original. Future modifications that resolve the above disadvantages are interesting directions for further research.

We studied only the case of supervised learning. Prior labels provided by the isolation forest can be used to apply ABIForest. However, this approach has an obstacle, namely, it is not clear how to process instances that have been incorrectly labeled by the isolation forest. These instances will contribute to incorrect results of ABIForest. There is another idea of developing an unsupervised ABIForest. By using an attention mechanism, we can increase the distance between the threshold for the path length and the real path length of each instance in each tree. Then, attention weights will increase the distance for a new instance in order to make a more robust decision about the instance. In this case, the attention weights will make the anomaly detection procedure more robust. However, this is a task for further research and for developing modifications of iForest based on an attention mechanism.

**Author Contributions:** Conceptualization, L.U. and A.K.; methodology, L.U. and V.M.; software, A.K. and A.A.; validation, A.A. and A.K.; formal analysis, A.A. and L.U.; investigation, A.K. and L.U.; resources, A.A. and V.M.; data curation, A.A.; writing—original draft preparation, L.U. and A.A.; writing—review and editing, A.K. and V.M.; visualization, A.A.; supervision, L.U.; project administration, V.M.; funding acquisition, V.M. All authors have read and agreed to the published version of the manuscript.

**Funding:** This research was partially funded by the Ministry of Science and Higher Education of the Russian Federation as part of the World-Class Research Center program: Advanced Digital Technologies (contract No. 075-15-2022-311, dated 20 April 2022).

**Acknowledgments:** The authors would like to express their appreciation for the anonymous referees, whose very valuable comments improved the paper.

**Institutional Review Board Statement:** Not applicable.

**Informed Consent Statement:** Not applicable.

**Data Availability Statement:** Not applicable.

**Conflicts of Interest:** The authors declare no conflict of interest.

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
