# Peer review of "Improved Anomaly Detection by Using the Attention-Based Isolation Forest"

_algorithms, doi:10.3390/a16010019_

Round 1

Reviewer 1 Report

Why F1 score is used as an evaluation metrics?

Include limitations of the study.

Include details about crossvalidation and hyperparameter tuning

Authors are suggested to include more discussion on the results and also include some explanation regarding the justification to support why the proposed method is better in comparison towards other methods

Reviewer 2 Report

This paper mainly studies the anomaly detection problem, and uses the isolated forest based on the attention mechanism for anomaly detection.

The main idea underlying the modification is to assign attention weights to each path of trees with learnable parameters depending on instances and trees themselves.

1.The improvements reported in the paper are small compared to original isolated forest algorithm in real dataset.In Figure 10 through 12, the improvements seem rather small. Why is such an improvement significant/important in practice?

2.What is the motivation for introducing the attention mechanism? What deficiencies do isolated forests have in anomaly detection that need to be improved through the attention mechanism?

3.Adding an overall flowchart of the method may help the reader better understand the architecture of the method

4.In this paper, the author mentioned that ABIForest is a supervised method. For unlabeled data, it is needed to determine priori labels by the original isolation forest. In the experiment, how much data is existing label data and the data whose prior labels are determined by the original isolation forest in the training.

5.Other types of methods are not compared between real datasets and synthetic datasets.The evaluation method is only the original isolated forest and ABIForest. Methods such as clustering, autoencoders, or other supervised methods can be added as baselines for comparison.

6.ABIForest has conducted a large number of ablation experiments on synthetic data sets, and the effect is very intuitive, clearly showing the performance improvement of ABIForest. However, only one type is compared in the real data set, and the author should select at least one of the six real data sets to repeat the above ablation experiment.

7.Figures 4 and 7 have wrong x-axis labels

Round 2

Reviewer 2 Report

The authors have solved most comments.

But the added Fig. 1 is  not shown in the modified manuscript in line 240.

general scheme of ABIForest is shown in Fig. ?”